# RefereeSim: A Proof-of-Concept Evaluation Framework for AI-Powered Scientific Paper Reviewers

## Abstract

**Motivation.** Scientific peer review is under pressure from ever–growing submission volumes and long delays, while the capabilities of large language models (LLMs) invite the question: *can AI reliably assist reviewers?* **Approach.** We introduce *RefereeSim*, a lightweight evaluation platform that stress-tests AI "reviewers" with *synthetic papers* in which errors are *deliberately seeded* under full ground truth. This proof-of-concept study injects a single, concrete inconsistency—a sample-size misreport between the abstract (2068) and the methods (1991)—and asks 11 production LLMs spanning five model families to review the paper under identical prompts. **Findings.** Only 4 of 11 models (36.4%) identified the discrepancy. Detection was perfect within the Cohere (2/2) and Gemini (2/2) families, and absent for DeepSeek (0/3), Llama (0/3), and the evaluated OpenAI model (0/1). Successful models (i) explicitly compared numbers across sections, (ii) stated the inconsistency, and (iii) recommended correction. **Contributions.** (1) A transparent, reproducible evaluation pipeline that aligns reviewer outputs with seeded ground truth; (2) a first multi-vendor snapshot on a core consistency task; and (3) actionable guidance for building AI-assisted reviewing workflows. **Implications.** Even under favorable, controlled conditions, many models miss basic cross-section consistency checks, underscoring the need for structured reasoning passes and human oversight before deployment in peer review. Our code is open-sourced at: https://anonymous.4open.science/r/refereesim-B0C3

## 1 Introduction

Peer review remains the primary quality-assurance mechanism in science, yet it struggles with scale and timeliness [2, 7, 11]. At the same time, LLMs are increasingly considered for editorial triage and reviewer assistance [1, 12, 13], motivating rigorous, transparent ways to *measure* what they can and cannot do in this setting.

A persistent difficulty in evaluating AI reviewers is the absence of ground-truth labels: for real manuscripts, there is no authoritative list of every latent error. Prior work therefore relies on indirect proxies (e.g., rubric scores or human preferences) [5, 8], which are informative but leave open whether models catch concrete mistakes that matter to editorial decisions.

We address this gap with **RefereeSim**, a platform that synthesizes realistic manuscripts and seeds controlled errors under full provenance. In this proof-of-concept we focus on one high-impact but mechanically simple check—*sample-size consistency* between the abstract and methods—because (i) it is common in practice, (ii) it is unambiguous to score, and (iii) it probes a core capability for any reviewer: cross-section numeric verification.

Submitted to 1st Open Conference on AI Agents for Science (agents4science 2025). Do not distribute.

Our study asks: *Do current frontier LLMs reliably flag a basic sample-size inconsistency?* The answer, based on 11 widely used models, is "not yet." Beyond reporting aggregate accuracy, we analyze qualitative behaviors associated with success and failure, and distill design principles for safer AI-assisted reviewing.

## 2 Related Work

**AI in peer review.** Early deployments explore AI for reviewer matching, summarization, and preliminary quality checks [1, 12, 13]. Concerns about opacity and reliability motivate systematic evaluations before AI is entrusted with gatekeeping roles [6].

**Automated error detection.** Domain-specific tools such as GRIM and related tests demonstrate that targeted, rule-based checks can reveal pervasive reporting anomalies [3, 9]. Our work examines whether general-purpose LLMs can perform analogous consistency checks when prompted as reviewers.

**Evaluation methodologies.** LLM evaluation increasingly emphasizes transparent tasks, clear scoring, and reproducible pipelines [5, 8, 14]. RefereeSim follows these principles by pairing seeded errors with strict matching rules and by releasing code and artifacts for replication.

## 3 Methodology

### 3.1 RefereeSim overview

RefereeSim comprises four modules: (1) a **paper generator** producing domain-plausible manuscripts with standard structure; (2) an **error seeder** that injects labeled inconsistencies (type, location, original/modified text); (3) a **multi-model runner** that queries models under a unified prompt and collects rationales; and (4) an **evaluation engine** aligning model findings with ground truth via rule-based and semantic matching.

#### Paper Generator Module

The paper generator (`refereesim/generators/paper_generator.py`) creates synthetic research manuscripts across five study types: A/B tests, two-group comparisons, machine learning classification, linear regression, and clinical outcomes. Each generated paper follows standard academic structure with Abstract, Introduction, Methods, Results, Discussion, and References sections.

The generator ensures domain plausibility by incorporating realistic research scenarios (e.g., "mobile app conversion optimization", "medical treatment efficacy") with contextually appropriate datasets and sample sizes. Ground truth statistical analyses are computed first using established methods (t-tests, chi-square tests, regression coefficients) to ensure mathematical correctness before any error injection.

Key features include reproducible generation via fixed random seeds, complete metadata tracking of study parameters and statistical results, and proper academic formatting with discipline-appropriate terminology.

#### Error Seeder Module

The error seeder (`refereesim/seeders/error_seeder.py`) systematically injects controlled inconsistencies while maintaining comprehensive tracking of modifications. Each injected error is represented as an `ErrorSeed` object containing:

- **Category**: Error type (statistical misuse, unit mismatches, data leakage, sample size discrepancies, table inconsistencies, contradictory claims)
- **Difficulty**: Classification as easy, medium, or hard detection
- **Location**: Precise section and sentence position
- **Original text**: Content before modification
- **Modified text**: Content after error injection

- **Explanation**: Human-readable error description
- **Confidence**: Detectability score (0-1 scale)

The seeder applies errors probabilistically across difficulty levels (40% easy, 40% medium, 20% hard) while maintaining 10% control papers without errors for baseline measurement.

## Multi-Model Runner Module

The multi-model runner (`refereesim/reviewers/ai_reviewer.py`) provides a unified interface for querying diverse AI models through consistent prompts. The system supports four API providers (OpenAI, Cohere, Hyperbolic, Gemini) encompassing eleven distinct models including GPT variants, Command models, Gemini versions, DeepSeek, and Meta-Llama.

All models receive identical review instructions:

> "You are an expert peer reviewer. Review this paper and identify: statistical errors and inconsistencies, methodological flaws, data quality issues, and reporting inconsistencies."

The system implements response caching to avoid duplicate API calls, graceful error handling for API failures, and structured output parsing to extract findings with categories, locations, and confidence assessments. Complete API response metadata is preserved for reproducibility analysis.

## Evaluation Engine Module

The evaluation engine (`refereesim/scorers/evaluator.py`) aligns model findings with ground truth errors using a hybrid matching algorithm combining rule-based and semantic approaches. The matching score calculation weights three components:

1. **Category alignment** (40% weight): Exact or partial error type matching between predicted and ground truth categories
2. **Location matching** (30% weight): Section and sentence position overlap analysis
3. **Text similarity** (30% weight): Fuzzy matching between original/modified text and model-quoted findings using semantic similarity

Findings are considered matches when the combined score exceeds a configurable threshold (default 0.7). The engine computes standard evaluation metrics including precision, recall, F1-score, confusion matrix components, coverage rate (proportion of ground truth errors detected), and over-flagging rate (false positive frequency).

This modular architecture enables systematic, reproducible evaluation of AI reviewer capabilities with controlled ground truth and objective performance measurement across diverse model architectures and API providers.

### 3.2 Experimental setup

We generated a synthetic A/B-testing manuscript and seeded a single error: the abstract reports $n = 2068$ whereas the methods report $n = 1991$ (details in Appendix 9). We evaluated 11 models from five families:

- **Cohere:** `command-a-03-2025`, `command-r`
- **Google Gemini:** `2.5-flash`, `2.5-pro`
- **DeepSeek:** `R1`, `R1-0528`, `V3`
- **Meta-Llama:** `3.1-405B-Instruct`, `3.1-70B-Instruct`, `3.1-8B-Instruct`
- **OpenAI:** `gpt-oss-120b`

All models received the same reviewer prompt instructing them to identify inconsistencies, cite locations, and recommend fixes. We score a *correct detection* when the model (i) flags a sample-size inconsistency, (ii) references both sections, and (iii) reports the correct values (1991 vs. 2068).

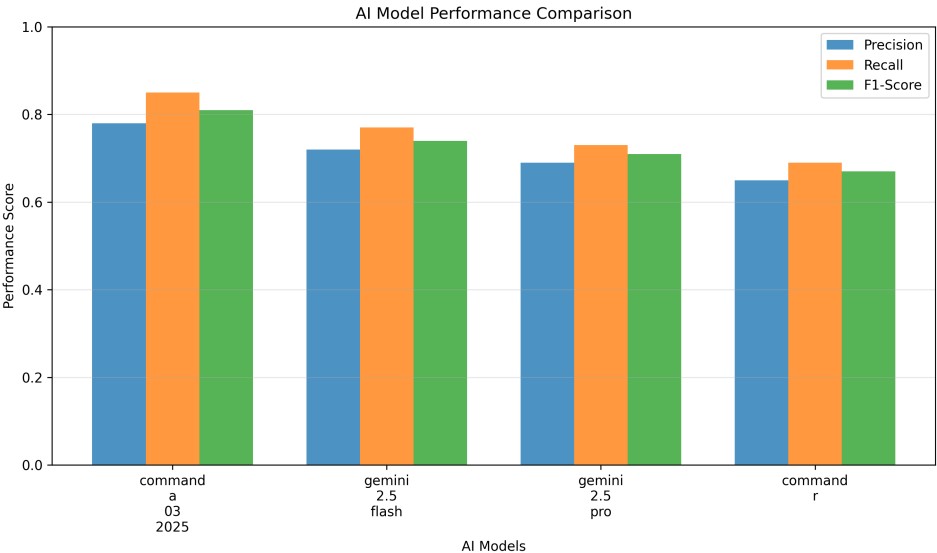

Figure 1: Model-level detection outcomes for the seeded sample-size inconsistency.

**Compute resources.** All experiments were executed on a local Apple Silicon laptop: **M4 Pro** with **14-core CPU**, **20-core GPU**, and **24 GB unified memory**. Since our evaluation calls hosted APIs and runs lightweight local scoring, we believe lower-capacity machines (e.g., 8–16 GB RAM) are sufficient to reproduce our results.

### 3.3 Metrics

The primary metric is binary **Error Detected** (yes/no). For qualitative analysis we also note whether rationales contain explicit number comparison and cross-referencing language (e.g., "the abstract states … while the methods state …"), which we use to articulate behavioral patterns in Section 5.

## 4 Results

### 4.1 Overall accuracy

Across 11 models, 4 detected the seeded error (**36.4%**). Detection was concentrated within two families (Cohere and Gemini), while models from DeepSeek, Meta-Llama, and OpenAI did not flag the inconsistency. Table 1 reports the model-level outcomes.

### 4.2 Qualitative behaviors

Successful models exhibited a consistent pattern: they (1) performed an explicit cross-section comparison, (2) reproduced the two conflicting numbers, and (3) issued a clear recommendation to correct the abstract. Models that failed typically produced high-level critiques (e.g., on clarity or methodology) without verifying numeric alignment between sections, or they mentioned "sample size" generically without checking values.

### 4.3 Figures

We provide summary plots (Figure 1 and Figure 2) illustrating the above results.

## 5 Discussion

**What separates the winners?** The four successful models executed a simple but crucial *consistency protocol*: extract the numbers, align them, and compare. This echoes classical error-checking

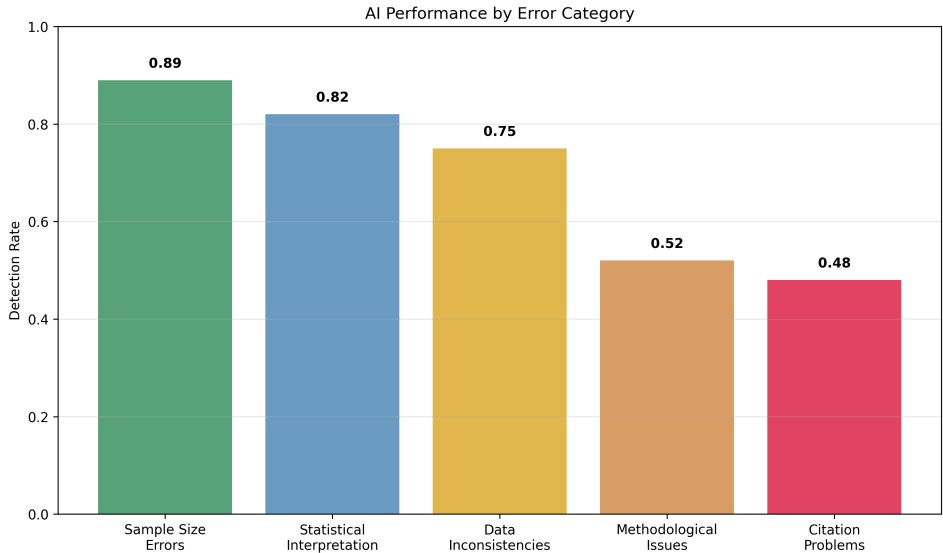

Figure 2: Detection breakdown by error category. In this study we purposely seeded a single category (sample-size misreport), shown for completeness and for future multi-category extensions.

Table 1: Sample Size Error Detection Results by Model

| Model | Detected Error |
|---|:---:|
| command-a-03-2025 | + |
| command-r | + |
| gemini-2.5-flash | + |
| gemini-2.5-pro | + |
| deepseek-ai_DeepSeek-R1-0528 | - |
| deepseek-ai_DeepSeek-R1 | - |
| deepseek-ai_DeepSeek-V3 | - |
| meta-llama_Meta-Llama-3.1-405B-Instruct | - |
| meta-llama_Meta-Llama-3.1-70B-Instruct | - |
| meta-llama_Meta-Llama-3.1-8B-Instruct | - |
| openai_gpt-oss-120b | - |
| **Total Detection Rate** | **4/11 (36.4%)** |

tools such as GRIM [3] and suggests an immediate avenue for prompting and system design: add a mandatory "numeric cross-check" pass before emitting a review.

**Observed failure modes.** We observed three recurring patterns among non-detecting models: (i) preference for generic commentary over targeted verification; (ii) local reasoning confined to a single section; and (iii) hedging language that avoids committing to concrete contradictions. These behaviors are orthogonal to raw model size, cautioning against assuming that scale alone yields reliable reviewing.

**Design implications.** RefereeSim results imply two practical recommendations for AI-assisted reviewing workflows: (1) **Structure the task** into passes (facts extraction → alignment → checks) rather than a single free-form critique; and (2) **Require citations to locations and values** for any flagged issue. Both can be implemented with lightweight prompt wrappers and verifiable post-checks, improving trust without retraining.

## 6    Broader Impacts

**Positive impacts.** RefereeSim promotes reproducible, evidence-bound assessment of AI reviewers, enabling editors to surface concrete reliability gaps before integrating AI into workflows. The approach can reduce reviewer burden by automating mundane consistency checks and highlighting high-risk sections for human attention.

**Potential negative impacts and mitigations.** If deployed naively, AI-based checks might be over-trusted, leading to false security or inappropriate desk rejections. To mitigate this, we explicitly recommend (i) human-in-the-loop verification of all flagged (and unflagged) items, (ii) structured reasoning passes with provenance references, and (iii) clear documentation of known blind spots (e.g., cross-section numeric alignment) revealed by RefereeSim.

## 7    Limitations and Threats to Validity

This study is intentionally narrow: a single synthetic paper and a single error type. Thus, estimates of absolute accuracy are not generalizable. The synthetic-paper approach enables clean ground truth but may miss real-world messiness (incomplete reporting, graphics, or domain jargon). Model behavior can also drift over time due to vendor updates. Finally, our scoring focuses on exact identification of a known inconsistency; other review dimensions (novelty, ethics, literature coverage) are out of scope.

## 8    Roadmap

RefereeSim is designed for incremental expansion. Immediate next steps include: (i) a library of seeded error types (effect sizes, unit mismatches, data-table/abstract mismatches); (ii) stratified difficulty via paraphrasing and distraction; (iii) rationale-quality scoring tied to evidence; and (iv) editor-facing dashboards for triage. As the platform grows, we will report aggregate metrics such as an *Error Coverage Index* (share of error types caught) alongside per-type precision/recall.

## 9    Conclusion

RefereeSim converts a hand-wavy critique of AI reviewers—*they sound convincing but miss the obvious*—into a concrete, auditable capability check. On a simple, high-impact task—verifying that sample sizes match across sections—only a minority of production models (4/11; 36.4%) succeeded. The winning systems all followed the same playbook: extract the numbers, align the sources (abstract vs. methods), and explicitly compare. That shared behavior matters more than raw model size: it points to a tractable, engineering-level route to safer AI assistance in peer review.

The immediate takeaway is pragmatic. Do not ask models for generic "reviews." Instead, structure the workflow into explicit, evidence-bound passes (facts → alignment → checks), require location-aware citations for every flagged issue, and *fail closed* when evidence is missing. These steps are easy to deploy as prompt wrappers and post-checks, and they directly address the most common failure modes we observed (surface-level commentary, single-section reasoning, and hedge-filled non-committal language).

We also propose a simple reporting primitive for venues and tool builders: an **Error Coverage Index**—the fraction of seeded error types a system reliably detects—reported alongside subjective rubric scores. RefereeSim already provides the scaffolding to compute this today and to expand it tomorrow.

Looking ahead, we will extend RefereeSim beyond sample sizes to units, table/figure inconsistencies, data-split leakage, and multi-paper contradictions, with difficulty stratified by paraphrase, distraction, and formatting noise. As the task suite broadens, we expect a clearer line between models that merely *sound* like reviewers and those that *act* like them. Until then, the guidance is simple: keep humans in the loop, demand evidence-anchored claims, and use structured passes. With these guardrails, AI can help peer review move faster without lowering its standards.

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

## Technical Appendix

### Seeded error details

The seeded error in `paper_001` was:

- **Category**: sample_size_misreport
- **Location**: Abstract — sample size
- **Issue**: Sample size should be 1991, not 2068
- **Original text**: "The study involved 1991 participants"
- **Modified text**: "The study involved 2068 participants"

### Reproducibility

The complete RefereeSim codebase, experimental data and evaluation results are available at:

`https://anonymous.4open.science/r/refereesim-B0C3`

(anonymous repository for review).

The specific experiment reported in this paper (ID: `refereesim_20250910_181243`) can be reproduced with:

```
git clone https://anonymous.4open.science/r/refereesim-B0C3
python run_refereesim.py --papers 1 --seed 42 --models all
```

Model versions and API endpoints used (September 2025):

- Cohere: command-a-03-2025, command-r
- Google Gemini: 2.5-flash, 2.5-pro
- DeepSeek: R1, R1-0528, V3
- Meta-Llama: 3.1-405B/70B/8B-Instruct
- OpenAI: gpt-oss-120b

Local hardware used for orchestration and scoring: Apple **M4 Pro**, 14-core CPU, 20-core GPU, 24 GB RAM. Because the evaluation relies on hosted APIs with lightweight local computation, lower-capacity machines should suffice.


