# Reproducibility Statement - Submission (#101)

RefereeSim: A Proof-of-Concept Evaluation
Framework for AI-Powered Scientific Paper
Reviewers

We release the complete RefereeSim codebase [1], the synthetic paper and seeded-error metadata, the exact review prompts, the evaluation scripts, and results at the anonymized artifact link provided in the paper (Technical Appendix – Seeded error details, Reproducivility, page 9).

The experiment reported (ID: refereesim 20250910 181243) can be reproduced end-to-end by cloning the repository

git clone https://anonymous.4open.science/r/refereesim-B0C3

or downloading the repository

https://anonymous.4open.science/r/refereesim-B0C3

and running python run_refereesim.py --papers 1 --seed 42 --models all.

Randomness is controlled via fixed seeds; all generated artifacts, prompts, and API inputs/outputs are logged with provider, model name/version, and timestamps, and responses are cached to enable exact re-computation of scores without re-querying APIs.

The matching rules used by the evaluation engine (category/location/text weights 0.4/0.3/0.3; match threshold 0.7) are implemented in code and documented.

We enumerate the model families and versions used as of September 2025 and include the saved outputs for the reported run to guard against vendor drift.

All results were produced on an Apple M4 Pro laptop (14-core CPU, 20-core GPU, 24 GB RAM); because inference is via hosted APIs, reproduction requires only modest local resources.