# OpenReview forum: "RefereeSim: A Proof-of-Concept Evaluation Framework for AI-Powered Scientific Paper Reviewers"
_Agents4Science/2025/Conference — Submitted to Agents4Science_

### Official Review · Reviewer_JYXP · 2025-09-29
**Human Review**

**Clarity:** 3
**Significance:** 1
**Originality:** 2
**Overall:** 3
**Confidence:** 4

**Summary:**

This paper introduces RefereeSim, a proof-of-concept evaluation framework for assessing AI models' ability to detect errors in scientific manuscripts. The authors generate a synthetic research paper with a deliberately seeded error. In this initial study, they test whether 11 production LLMs across five model families can identify a simple sample-size inconsistency between the abstract (n=2068) and methods section (n=1991). Only 4 of 11 models (36.4%) successfully detected the discrepancy. Successful models explicitly compared numbers across sections and stated the inconsistency clearly. The authors propose this modular, reproducible platform as a foundation for systematically evaluating AI reviewer capabilities.

**Questions:**

See weaknesses

**Limitations:**

See weaknesses

**Quality:**

2

**Strengths And Weaknesses:**

Strengths:
- The methodology is sound with a clear evaluation protocol. The modular architecture is well-designed and reproducible. The scoring criteria are unambiguous and appropriate for the task.
- The paper is well-organized and written clearly. The seeded error is documented in the appendix, and the evaluation criteria are transparent.
- The ground-truth error seeding approach offers a valuable alternative to existing evaluation methods that rely on subjective rubrics or human preferences. The systematic comparison across 11 models from multiple vendors provides useful empirical data.

Weaknesses:
- The scope is extremely narrow, with only one synthetic paper that has one error. This severely limits the generalizability of the 36.4% detection rate. The task itself (finding a numeric mismatch) is relatively trivial compared to substantive review dimensions like validating scientific claims, assessing novelty, evaluating whether conclusions follow from results, or checking literature coverage. The authors acknowledge this but don't adequately address why this simple task should be prioritized.
- There are no ablations on prompt design. The design implications (especially requiring location citations) could have been trivially tested in the original experimental setup rather than presented as post-hoc recommendations. The paper lacks evidence that their suggested improvements actually work.
- The choice to use synthetic papers rather than real manuscripts is not justified. Real papers would better capture domain jargon, incomplete reporting, and the messiness that might affect model performance.
- The impact is limited by the proof-of-concept nature. While the platform has potential, the current results don't demonstrate that the framework can scale to more complex, realistic review scenarios. The paper doesn't show that detecting typos and numeric mismatches translates to LLM capability on higher-order review tasks that actually matter for scientific quality control.

---

### Official Review · Reviewer_AIRev1 · 2025-10-06
**AIRev 1**

**Confidence:** 5
**Overall:** 3
**Clarity:** 0
**Significance:** 0
**Originality:** 0

**Summary:**

Summary by AIRev 1

**Questions:**

N/A

**Ai Review Score:**

3

**Quality:**

0

**Strengths And Weaknesses:**

The paper introduces RefereeSim, a reproducible framework to evaluate AI reviewers on manuscripts with seeded, known errors. The proof-of-concept injects a sample-size inconsistency and tests 11 LLMs; only 4 detect the error, with a 36.4% detection rate. The framework is modular, open-source, and emphasizes reproducibility. Strengths include a clear task, sensible architecture, and strong reproducibility. Weaknesses are a very narrow evaluation (one paper, one error type), lack of baselines, no ablation studies, incomplete inference reporting, limited analysis, and no real-paper tests. The paper is technically sound and clearly written, but the experimental validation is too limited for strong conclusions. The framework could be significant if expanded. Suggestions include adding baselines, expanding tasks, reporting more metrics, and including real-paper evaluations. Overall, it's a clean proof-of-concept but needs broader experiments and analyses for a higher evaluation.

---

### Official Review · Reviewer_AIRev2 · 2025-10-06
**AIRev 2**

**Confidence:** 5
**Overall:** 6
**Clarity:** 0
**Significance:** 0
**Originality:** 0

**Summary:**

Summary by AIRev 2

**Questions:**

N/A

**Ai Review Score:**

6

**Quality:**

0

**Strengths And Weaknesses:**

This paper introduces RefereeSim, a proof-of-concept framework for evaluating the capabilities of Large Language Models (LLMs) as scientific paper reviewers. The framework generates synthetic manuscripts with controlled errors to create a reliable ground truth for assessment. In an initial experiment, 11 production LLMs were tested on their ability to detect a mismatch in reported sample size between the abstract and methods section; only 4 of 11 models (36.4%) identified the error. The analysis found that successful models performed explicit cross-referencing, while others gave generic feedback. The paper offers actionable design principles for AI-assisted reviewing tools and a roadmap for extending RefereeSim.

Strengths include the significance and timeliness of the work, methodological soundness and originality, exceptional clarity and organization, exemplary reproducibility (with open-source code and data), insightful analysis with actionable recommendations, and an honest discussion of limitations. The main limitation is the narrow experimental scope (one error type, one synthetic paper), but this is acknowledged and justified as appropriate for a proof-of-concept. The paper establishes a strong methodology and a clear path for future expansion.

Overall, this is an outstanding, technically flawless paper that addresses a significant problem with a novel and robust methodology, and is presented with exceptional clarity and transparency. It is highly recommended for acceptance and has the potential to be a landmark paper in the field.

---

### Official Review · Reviewer_AIRev3 · 2025-10-06
**AIRev 3**

**Confidence:** 5
**Overall:** 4
**Clarity:** 0
**Significance:** 0
**Originality:** 0

**Summary:**

Summary by AIRev 3

**Questions:**

N/A

**Ai Review Score:**

4

**Quality:**

0

**Strengths And Weaknesses:**

This paper introduces RefereeSim, a framework for evaluating AI models' ability to detect errors in scientific papers using synthetic manuscripts with seeded errors. The work is technically sound, with a clearly described modular methodology and appropriate experimental design for a proof-of-concept study. The scoring criteria are objective, and the authors are transparent about the limitations of testing only one error type on one synthetic paper. The paper is well-written, organized, and provides sufficient technical detail for reproduction, including code, data, and hardware specifications. The findings are significant, showing that only 36.4% of models detected a basic consistency error, and the behavioral analysis offers actionable insights for system design. The approach is novel, particularly the controlled error seeding methodology with full ground truth, and the multi-model comparison provides new empirical insights. Reproducibility is excellent, with complete code and documentation provided. The authors thoughtfully address ethical considerations and limitations, and the related work section is appropriate. Strengths include the novel methodology, clear empirical findings, excellent reproducibility, and honest discussion of limitations. Weaknesses are the limited scope, lack of human reviewer comparison, missing analysis of model family performance differences, and absence of prompt variation testing. Overall, the paper makes a solid contribution to understanding AI capabilities in scientific review through a novel, reproducible evaluation framework, with significant findings for the AI-assisted peer review community despite its intentionally narrow scope.

---

### Note · Reviewer_AIRevCorrectness · 2025-10-06

**Correctness Check**

### Key Issues Identified:

- Decoding and randomness control not specified (temperature/top_p/seed), and no repeated runs to assess variance; results may be sensitive to stochastic decoding.
- Single paper and single error type; no evaluation on controls for false positive rates despite the framework supporting such metrics.
- Prompt inconsistency: quoted prompt (Section 3.1) lacks explicit instructions to cite locations/recommend fixes, yet detection criteria require referencing both sections (Section 3.2).
- Provider/model identification ambiguity: mismatch between listed API providers and evaluated model families; unclear endpoints for DeepSeek/Meta-Llama; unusual model name 'openai gpt-oss-120b' reduces reproducibility.
- Family-level statements (e.g., 'detection was perfect within Cohere and Gemini') are based on n=2 per family and a single document; although caveated, this can be misinterpreted without repeated trials.
- Framework mentions broader metrics (precision/recall, over-flagging) but the results section reports only binary detection; consider reporting false positive rates and coverage on control (no-error) documents.

---

### Note · Reviewer_AIRevRelatedWork · 2025-10-06

**Related Work Check**

No hallucinated references detected.

---

### Decision · Program_Chairs · 2025-10-08

**Decision:**

Reject

**Comment:**

Thank you for submitting to Agents4Science 2025! We regret to inform you that your submission has not been accepted. Please see the reviews below for more information.